# Deficits in Face Recognition and Consequent Quality-of-Life Factors in Individuals with Cerebral Visual Impairment

**Corinna M. Bauer** [1,2,*], **Claire E. Manley** [2], **John Ravenscroft** [3], **Howard Cabral** [4], **Daniel D. Dilks** [5] **and Peter J. Bex** [6]

1 The Lab of Neuroimaging and Vision Science, Gordon Center for Medical Imaging, Department of Radiology, Massachusetts General Hospital, Harvard Medical School, Boston, MA 02114, USA
2 Department of Ophthalmology, Massachusetts Eye and Ear, Harvard Medical School, Boston, MA 02114, USA
3 Scottish Sensory Centre, Moray House School of Education and Sport, University of Edinburgh, Edinburgh EH8 8AQ, UK
4 Department of Biostatistics, Boston University School of Public Health, Boston, MA 02118, USA
5 Dilks Lab, Department of Psychology, Emory University, Atlanta, GA 30322, USA
6 Translational Vision Lab, Department of Psychology, Northeastern University, Boston, MA 02115, USA
* Correspondence: cbauer@mgh.harvard.edu

**Abstract:** Individuals with cerebral visual impairment (CVI) frequently report challenges with face recognition, and subsequent difficulties with social interactions. However, there is limited empirical evidence supporting poor face recognition in individuals with CVI and the potential impact on social–emotional quality-of-life factors. Moreover, it is unclear whether any difficulties with face recognition represent a broader ventral stream dysfunction. In this web-based study, data from a face recognition task, a glass pattern detection task, and the Strengths and Difficulties Questionnaire (SDQ) were analyzed from 16 participants with CVI and 25 controls. In addition, participants completed a subset of questions from the CVI Inventory to provide a self-report of potential areas of visual perception that participants found challenging. The results demonstrate a significant impairment in the performance of a face recognition task in participants with CVI compared to controls, which was not observed for the glass pattern task. Specifically, we observed a significant increase in threshold, reduction in the proportion correct, and an increase in response time for the faces, but not for the glass pattern task. Participants with CVI also reported a significant increase in sub-scores of the SDQ for emotional problems and internalizing scores after adjusting for the potential confounding effects of age. Finally, individuals with CVI also reported a greater number of difficulties on items from the CVI Inventory, specifically the five questions and those related to face and object recognition. Together, these results indicate that individuals with CVI may demonstrate significant difficulties with face recognition, which may be linked to quality-of-life factors. This evidence suggests that targeted evaluations of face recognition are warranted in all individuals with CVI, regardless of their age.

**Keywords:** cerebral visual impairment (CVI); cortical visual impairment (CVI); face recognition; ventral stream dysfunction; Strengths and Difficulties Questionnaire; CVI inventory; visual perception; glass pattern

## 1. Introduction

Ventral visual functions form the bedrock of many of our essential everyday behaviors, including face recognition, place recognition, object recognition, reading, and social cognition [1,2]. As such, impairments in any of these functions have potential widespread consequences for a person's independence and participation in vocational and social functions [3]. For example, the inability to recognize faces can cause fear, anxiety, and difficulty with social interactions, at times causing those with so-called prosopagnosia to avoid social situations [4,5]. The extent to which social and emotional well-being are impacted

in individuals with face recognition difficulties due to cerebral visual impairment (CVI) is unknown.

Cerebral visual impairment is the foremost individual cause of pediatric visual impairment in high income countries and its incidence is on the rise in lower- and middle-income countries [6–14]. CVI is an umbrella term encompassing a variety of visual dysfunctions as a consequence of brain injury, malformation, and/or genetic disorders [15]. Generally speaking, children with CVI may be placed into one of four profiles based on the severity of functional impact [16]; in each profile, the extent of the visual dysfunctions are beyond what may be expected due to any potentially co-occurring ocular visual impairment [17]. While many individuals with CVI demonstrate dorsal stream dysfunctions, such as impairments in visual attention and motion perception [18–20], many may also exhibit ventral stream dysfunctions. For example, difficulties with face recognition are regularly described in anecdotal reports of children with CVI [21–23]. Indeed, parents of children with CVI frequently report that their child cannot recognize them when they pick them up from school or even in family photographs—yet beyond these subjective reports, these impairments are poorly understood. It is also unclear how these impairments contribute to mental health and well-being in individuals with CVI.

Thus, the current study sought to address these gaps in our knowledge by investigating three main research questions: 1. to empirically investigate face recognition abilities in individuals with CVI using a forced four-choice match to sample paradigm whereby the target was presented with varying degrees of Gaussian blur; 2. to determine whether other ventral stream functions are also impacted by using a static glass-pattern-form detection task; and 3. to consider the potential impact of face recognition difficulties on social and emotional metrics of quality of life based on subscales of the Strengths and Difficulties Questionnaire (SDQ). The results from this study indicate that CVI may be associated with significant impairments in face recognition that negatively impact quality-of-life factors.

## 2. Methods

### 2.1. Participants

Participants were invited to partake in this online study through targeted email, social media advertisements, and as part of ongoing research studies. A total of 53 individuals participated in the study between the ages of 9.44 and 71.64. To help mitigate the potential decline in face recognition associated with aging, data from 12 participants were excluded because they were over the age of 35 years. Thus, data from 41 participants (16 CVI, 25 controls) are represented in these analyses. To date, 40 participants completed the glass pattern and face recognition tasks (97.56%), 36 completed the SDQ (87.80%), and 37 completed a subset of items from the CVI Inventory [24] (92.68%). The presence of CVI was noted based on a participant report for online-only participants and, for those recruited in person, the diagnosis of CVI was confirmed by an eye care professional based on their clinical assessment including visual function and functional vision. Specifically, CVI was diagnosed when the participant had a suspect medical history and where their level of functional vision was worse than could be expected based on any potentially co-occurring ocular impairment. Participant details can be found in Table 1.

Participants provided informed consent prior to data collection. This study was approved by the Investigative Review Board at Northeastern University and was carried out in accordance with the Code of Ethics of the World Medical Association (Declaration of Helsinki) for experiments involving humans.

**Table 1.** Participant demographic information for the CVI and control groups. Group mean, standard deviation (s.d.), and range values are also provided for the face recognition task, glass pattern task, Strengths and Difficulties Questionnaire, and CVI Inventory. Where appropriate T- or F-statistics and associated *p*-values are provided.

| | *n* CVI | CVI Mean (s.d.) | CVI Min | CVI Max | *n* Control | Control Mean | Control Min | Control Max | T/F Stat | *p*-Value | adj. *p*-Value |
|---|---|---|---|---|---|---|---|---|---|---|---|
| Age | 16 | 18.78 (6.51) | 8.56 | 30.36 | 25 | 20.99 (4.82) | 9.44 | 34.6 | −1.24 | 0.2209 | |
| Sex | | 15 female | | | | 15 female | | | 0.026 | 0.87 | |
| **Face Recognition Task** | | | | | | | | | | | |
| Face threshold | 14 | 51.93 (45.36) | 11.02 | 148.77 | 22 | 16.70 (6.15) | 3.32 | 30.65 | 3.01 | 0.0002 | 0.0006 |
| Face response time (ms) | | 3078.50 (388.88) | 2593.90 | 3658.35 | | 2744.70 (385.87) | 1945.05 | 3294.28 | 12.22 | 0.011 | 0.032 |
| Face proportion correct | | 0.56 (0.16) | 0.18 | 0.83 | | 0.75 (0.08) | 0.63 | 0.88 | 38.14 | <0.0001 | <0.0001 |
| Face *n* null | | 3 (2.78) | 0 | 10 | | 1.41 (1.14) | 0 | 3 | 2.1 | 0.051 | |
| **Glass Pattern Task** | | | | | | | | | | | |
| Glass threshold | 13 | 0.52 (0.20) | 0.29 | 1.00 | 25 | 0.45 (0.15) | 0.05 | 0.67 | 1.66 | 0.30 | 0.90 |
| Glass response time (ms) | | 2368.12 (646.56) | 1646.80 | 3836.05 | | 2461.20 (523.95) | 1413.05 | 3310.25 | 0.05 | 0.65 | 1 |
| Glass proportion correct | | 0.5 (0.15) | 0.23 | 0.70 | | 0.55 (0.11) | 0.38 | 0.78 | 3.06 | 0.26 | 0.79 |
| Glass *n* null | | 3.87 (4.52) | 0 | 18 | | 3 (2.57) | 1 | 11 | 0.68 | 0.51 | |
| **Strengths and Difficulties Questionnaire** | | | | | | | | | | | |
| Emotional problems scale | 14 | 5.21 (2.42) | 2 | 10 | 21 | 3.14 (1.90) | 0 | 6 | 8.83 | 0.0056 | 0.045 |
| Conduct problems scale | | 1.86 (1.41) | 0 | 5 | | 1.43 (1.08) | 0 | 4 | 1.5 | 0.23 | 1 |
| Hyperactivity scale | | 3.57 (2.38) | 0 | 8 | | 3.24 (2.10) | 0 | 7 | 0.17 | 0.68 | 1 |
| Peer problems scale | | 3 (1.71) | 1 | 7 | | 1.86 (1.31) | 0 | 6 | 4.4 | 0.044 | 0.35 |
| Prosocial scale | | 8 (1.96) | 5 | 10 | | 8.71 (1.45) | 4 | 10 | 1.11 | 0.30 | 1 |
| Total difficulties score | | 13.64 (4.70) | 8 | 22 | | 9.67 (4.62) | 2 | 19 | 6.38 | 0.017 | 0.13 |
| Externalizing score | | 5.43 (2.87) | 1 | 10 | | 4.67 (2.74) | 1 | 11 | 0.72 | 0.40 | 1 |
| Internalizing score | | 8.21 (3.51) | 4 | 14 | | 5 (2.35) | 1 | 8 | 10.7 | 0.0026 | 0.021 |
| **CVI Inventory** | | | | | | | | | | | |
| Faces positive screen | 15 | 2 (1.41) | 0 | 4 | 21 | 0.19 (0.51) | 0 | 2 | 32.07 | <0.0001 | <0.0001 |
| Objects positive screen | | 1.8 (1.26) | 0 | 4 | | 0 (0) | 0 | 0 | 46.67 | <0.0001 | <0.0001 |
| The Five Questions positive screen | | 3.73 (1.49) | 0 | 5 | | 0.57 (0.87) | 0 | 3 | 65.93 | <0.0001 | <0.0001 |

## 2.2. Face Recognition Task

A forced four-choice match to sample design was used to evaluate face recognition. In this task, participants were shown a target face with four faces beneath from which they were asked to identify the match using PsyToolkit (https://www.psytoolkit.org/, accessed on 5 September 2022) [25,26] (see Figure 1 for an example stimulus). We report values in pixels because many of the participants completed the study online and we had no control over the display properties or viewing distance; participants were only

instructed to view the stimuli from a comfortable distance. Face stimuli were selected from the Interdisciplinary Affective Science Laboratory (IASLab) Face Set which comprises photographs of 50 individuals (31 female, 19 male) in multiple poses. We selected faces with straight-ahead gaze, closed mouth and with either neutral or calm expressions. The images were 600 pixels vertically $\times$ 400 pixels horizontally and we created an oval window of 240 pixels vertically $\times$ 160 pixels horizontally that was centered on each face, and removed background features and most of the hair that could provide non-face information. For each trial, we selected a target individual and 3 non-target individuals at random without replacement from the faces set. We selected 2 different images of the target individual and 1 image of each non-target individual. To remove luminance, contrast, and chrominance cues of the target identity, we averaged the RGB images of 5 faces. We then extracted the luminance and chrominance planes of the images using MATLAB's (The Mathworks, Inc., Natick, MA, USA) function rgb2ycbcr(). Next, we sorted the pixel luminance of the 5 faces and the average face in ascending order, then swapped pixels from the 5 faces with those of the average face in ascending order, resulting in all five faces having exactly the same pixel values and distribution, but in different locations in each face, and thus removing any luminance or contrast differences between faces. Next, we assigned the same chrominance to each face by replacing the $C_b$ and $C_r$ planes of each face with the $C_b$ and $C_r$ planes of the average face. We then used MATLAB's function ycbcr2rgb() to create 5 RGB faces with identical chrominance and luminance properties, thereby removing these cues as potential artefacts that could be used to identify the target face. In each trial, the target sample face was blurred with a Gaussian function with a standard deviation of 10 log-spaced steps between 1 pixel and 50 pixels (corresponding to 1/8th of the image width and 1/4 of the face width). Participants were shown a total of 40 trials, with 4 per level of difficulty in random order. The number of stimuli correctly identified as a function of blur was fit with a cumulative Gaussian function, from which the blur threshold was estimated at the 62.5% correct level. Blur threshold, response time, and proportion correct were measured. Each trial timed out after 6000 ms. If the participant did not respond within the time limit, it was counted as an incorrect response. The number of trials that timed out were recorded to differentiate between incorrect guesses and a null response.

*2.3. Glass Pattern Task*

For the glass pattern task, participants were asked to choose which stimulus (out of a choice of four) was embedded with a circular pattern. Stimuli were presented using PsyToolkit[1,2] (see Figure 2 for an example stimulus). Glass pattern stimuli were arranged in 4 quadrants, each 250 $\times$ 250 pixels, separated by a gap of 25 pixels. Each quadrant contained 200 white (RGB = [255 255 255]) dipoles (dot pairs) on a mean value background (RGB = [127 127 127]). One of the quadrants at random contained a circular glass pattern signal, and the other 3 quadrants contained only noise dipoles. The orientation of signal dipoles was tangential to the orientation of the mid-point of the dipole relative to the quadrant center; the orientation of noise dipoles was random. The proportion of signal dots was varied from 10% to 100% in 10 evenly spaced log steps. Participants were shown a total of 40 trials, with 4 per level of difficulty. The number of stimuli correctly detected (as a function of coherence) was fit with a cumulative Gaussian function, from which a coherence threshold was estimated at the 62.5% correct level. Better performance was denoted by a lower threshold, whereby a smaller percentage of the dots needed to be coherently oriented for the participant to detect the pattern. Coherence threshold, response time, and proportion correct were measured. Similar to the face recognition task, each trial timed out after 6000 ms. If the participant did not respond within the time limit, it was counted as an incorrect response. The number of trials that timed out were recorded to differentiate between incorrect guesses and a null response.

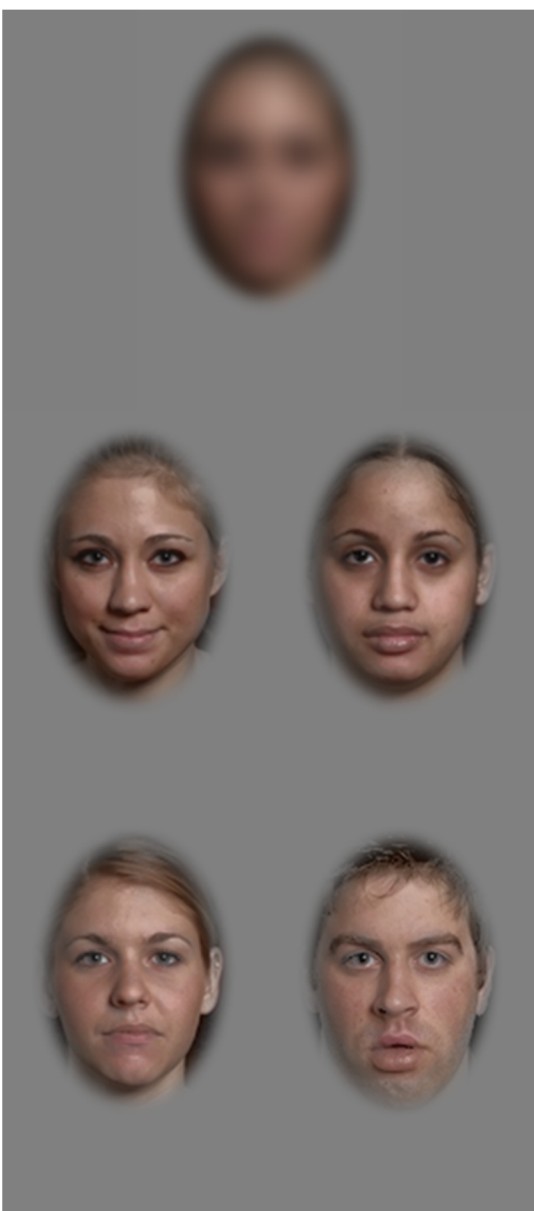

**Figure 1.** Example stimuli from the face recognition task. The top center image shows a blurred image ($\sigma$ = 13.6 pixels in this example) of the target individual. One of the lower 4 unblurred faces (chosen at random across trials) was a different image of the target individual, the other 3 faces were different individuals. Participants identified, using mouse click or screen touch, which of the lower 4 faces was the same individual as the top blurred face, guessing if necessary.

*2.4. CVI Inventory*

The CVI inventory [27] was designed as a structured history-taking tool to provide an understanding of how CVIs manifest in an individual with a diagnosis of (or at risk of) CVI. Multiple domains of vision are targeted including visual attention, ventral stream functions, dorsal stream functions, and visual function. A subset of five questions was shown to be particularly sensitive at differentiating CVI from other potential diagnoses [24,27,28]. These five questions, as well as those pertaining to ventral stream functions, such as face and object recognition, were administered as part of this online study. The specific questions are listed in the Supplementary Material. Each question was rated by the participant or their caregiver as being true: never, rarely, sometimes, or always. Within each subcategory (i.e., faces, objects, and the 5 questions), the number of responses categorized as sometimes

or always were summed for a maximum of 4 positive responses for faces, 5 for objects, and 5 for the "5 questions".

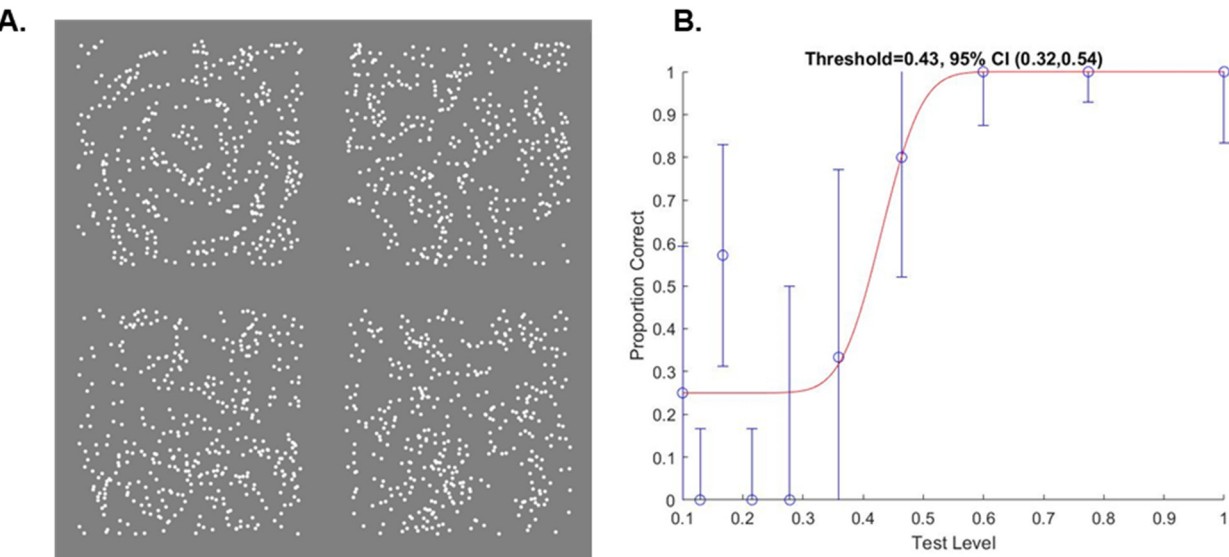

**Figure 2.** Example stimulus and psychometric function from the glass pattern task. (**A**) Each stimulus contained four quadrants, each of which contained 200 dipoles. The orientation of signal dipoles formed a global circular glass pattern (top left quadrant), and the orientation of noise dipoles was random. One of the 4 quadrants (at random across trials) contained a proportion (1.0 in this example) of signal and noise dipoles, and the remaining 3 quadrants contained only noise dipoles. Participants identified, using mouse click or screen touch, which of the 4 quadrants contained the circular pattern, guessing if necessary. (**B**) Typical psychometric function for one participant with CVI. The x axis shows the signal coherence (proportion of signal elements in the target quadrant), and the y axis shows the proportion of trials whereby the participant correctly identified the target quadrant. Circles show the proportion correct trials, error bars show the binomial standard deviation, and the red line shows the best fitting cumulative Gaussian function from which threshold and lower and upper 95% confidence intervals were estimated, shown in the title.

### 2.5. Strengths and Difficulties Questionnaire

The self-report version of the SDQ was administered. Participants were asked to indicate for each question whether the sentence was "Not True", "Somewhat True", or "Certainly True" based on their personal experiences. They were instructed to answer all items to the best of their ability based on their behavior over the last six months or during the current school year. Each item was graded according to the published scoring criteria and summed for a total difficulties score. Likewise, sub-scores for emotional problems, conduct problems, hyperactivity, peer problems, prosocial behavior, externalizing, and internalizing were also generated.

### 2.6. Statistical Analysis

Potential differences in age between groups were evaluated using Student's *t*-test, assuming unequal variance. The chi-square test was used to investigate potential group differences in the distribution of males and females. A *t*-test was used to compare the age distribution and frequency of null responses (i.e., timed out) between groups. Group differences in outcomes (blur/coherence threshold, response time and proportion correct) for the face recognition and glass pattern data were evaluated in two ways: 1. For those participants with valid responses in both tasks, we used a series of two (group) by two (task) repeated measure mixed model analyses with and without controlling for the potential effects of age. 2. To include the full sample of participants, we also investigated between-group differences in outcome measures using Mann–Whitney U or *t*-tests based on tests

of normality. Additionally, an analysis of covariance (ANCOVA) was used to evaluate between-group differences in outcomes while controlling for the potential impact of age. Group differences in the SDQ responses were investigated using the Mann–Whitney U, while the Spearman correlation was used to determine the relationship between SDQ, face recognition (blur/coherence threshold, response time, and proportion correct), and CVI Inventory scores. For each *t*-test, equality of variance between groups was evaluated and Satterthwaite estimates were used in instances of unequal variance. Multiple comparisons correction was performed using the Bonferroni method. For all analyses, $p < 0.05$ was chosen as the critical threshold for representing statistically significant effects.

All statistical analyses were performed using SAS OnDemand for Academics (SAS Institute, Inc., Cary, NC, USA).

## 3. Results

### 3.1. Participants

There were slightly more females ($n = 25$) than males ($n = 16$) overall, but this difference did not reach statistical significance (chi-square = 1.98, $p = 0.16$). There were also no significant differences in the distribution of males and females between groups (chi-square = 0.0256, $p = 0.87$). For the face task, data from two participants with CVI and three controls were removed as outliers (i.e., threshold exceeded three standard deviations beyond the mean) or due to negative threshold values. Data from the glass pattern task were removed from two participants with CVI whose thresholds were greater than one.

There was no significant difference in age between participants with CVI (age $_{CVI}$ = 18.78 years, 6.51 s.d., range = 8.56–30.36 years) and control participants (age $_{control}$ = 20.99 years, 4.82 s.d., range = 9.44–34.60; t(39) = −1.24, $p = 0.22$). There were also no significant correlations between age and outcome variables between groups or in the control group ($p > 0.1$). In the CVI group, there was a significant correlation between age and proportion correct for the faces task, but this did not survive correction for multiple comparisons (r = −0.61, $p = 0.015$, adj. $p = 0.089$).

There was no significant difference between groups for the number of null responses in the face task (i.e., time-out trials) (CVI $_{mean}$ = 2.88, 2.73 s.d.; control $_{mean}$ = 1.63, 1.58 s.d.; t(21.766) = 1.66, $p = 0.1121$). No group differences in null responses were found for the glass pattern task (CVI $_{mean}$ = 3.87, 4.52 s.d.; control $_{mean}$ = 3.00, 2.57 s.d.; t(19.75) = 0.68, $p = 0.5059$).

### 3.2. Repeated Measures Adjusting for Age

A total of 33 participants (CVI = 11, control = 22) completed both tasks and were included in the repeated measures analyses.

#### 3.2.1. Threshold

There was a significant main effect of task (F(1, 31) = 25.37, $p < 0.0001$) and group (F(1, 31) = 5.72, $p = 0.023$), but not of age (F(1, 31) = 0.41, $p = 0.53$), on threshold. There was also a significant interaction between task and group (F(1, 31) = 5.42, $p = 0.027$). Specifically, the CVI group showed a significant increase in threshold for the faces task compared to controls (t(31) = 3.34, $p = 0.0022$, adj. $p = 0.0088$), but not for the glass pattern task (t(31) = 0.03, $p = 0.98$, adj. $p = 1$). Refer to Figure 3 and Table 1 for additional details.

#### 3.2.2. Proportion Correct

There was a significant main effect of task (F(1, 31) = 24.21, $p < 0.0001$) and group (F(1, 31) = 5.26, $p = 0.029$) on proportion correct, but not on age (F(1, 31) = 0.03, $p = 0.86$). There was also a significant interaction between task and group (F(1, 31) = 6.33, $p = 0.017$). Specifically, CVI was associated with a significant reduction in the proportion correct compared to controls for the faces task (t(31) = −3.21, $p = 0.0031$, adj $p = 0.012$), but not for the glass pattern task (t(31) = −0.83, $p = 0.42$ adj $p = 1$).

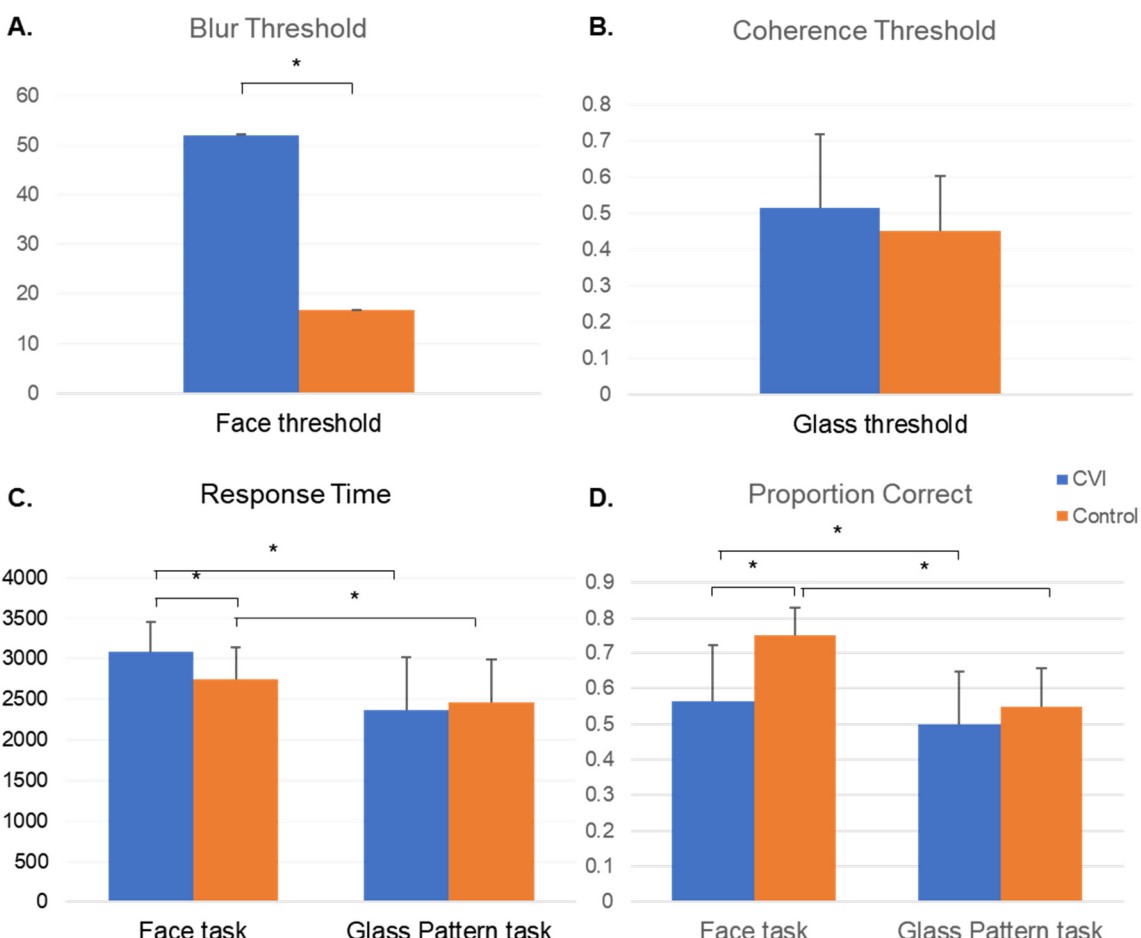

**Figure 3.** Mean (s.d.) results for (**A**) face threshold, (**B**) glass pattern threshold, (**C**) response time, and (**D**) proportion correct for the CVI (blue) and control (orange) groups. (**A**) The CVI group showed a significantly higher blur threshold for the face perception task, indicating that they were more susceptible to the effects of blur and required more clarity in order to perform the task and identify the matching face (i.e., higher threshold reflects worse performance). (**B**) Although the CVI group demonstrated a qualitatively higher threshold on the glass pattern task, this did not reach statistical significance ($p > 0.05$). (**C**) The CVI group showed an increased response time compared to controls for the faces task, but not for the glass pattern task. Within each group, participants took longer to perform the face recognition task compared to the glass pattern task. (**D**) The CVI group achieved a lower proportion correct on both tasks compared to controls, but this only reached statistical significance for the face recognition task ($p < 0.0001$). Within each group, participants achieved a higher proportion correct for the faces task compared to the glass pattern task ($p < 0.05$ in both groups). * indicates $p < 0.05$.

### 3.2.3. Response Time

There was a significant main effect of task ($F_{(1, 31)} = 64.42$, $p < 0.0001$), but not group ($F_{(1, 31)} = 0.03$, $p = 0.8727$), on response time. There was a significant interaction between task and group ($F_{(1, 31)} = 15.87$, $p = 0.0004$). Specifically, CVI was associated with an increase in response time for the faces task compared to controls ($t(31) = 1.73$, $p = 0.094$, adj. $p = 0.38$), although this did not reach statistical significance. There were also no significant group differences for the glass pattern task ($t(31) = -2.01$, $p = 0.053$, adj. $p = 0.21$).

The above analyses were repeated without adjusting for the potential impact of age. The results were virtually unchanged (see Supplementary Material for details).

*3.3. Non-Repeated Measures Covarying for Age*

3.3.1. Faces

Threshold

Following tests for normal distribution, the Mann–Whitney U revealed a significant increase in face threshold for the CVI group compared to controls (S = 362, Z = 3.33, $p = 0.0021$, exact $p = 0.0005$). These results remained significant after adjusting for the potential confounds of age (adjusted mean face threshold CVI = 54.69, adjusted mean face threshold control = 14.94; F(1, 35) = 17.50, $p = 0.0002$, adj. $p = 0.0006$, age (F(1, 35) = 4.53, $p = 0.41$)).

Proportion Correct

*T*-tests revealed that the CVI group had a significantly lower proportion correct on the face recognition task compared to controls (CVI $_{mean}$ = 0.56, 0.16 s.d., control $_{mean}$ = 0.75, 0.081 s.d.; t(17.38) = −4.05, $p = 0.0008$). These results remained significant after adjusting for age (adjusted mean proportion correct CVI = 0.55, adjusted mean proportion correct control = 0.76; F(1, 35) = 30.27, $p < 0.0001$, adj. $p < 0.0001$; age (F(1, 35) = 7.15, $p = 0.012$)).

Response Time

The CVI group had a significantly higher response time on the face recognition task compared to controls (CVI $_{mean}$ = 3078.5 ms, 388.9 s.d., control $_{mean}$ = 2744.7 ms, 385.9 s.d.; t(34) = 2.52, $p = 0.017$). This difference remained significant after adjusting for age (adjusted mean response time CVI = 3098.74 ms, adjusted mean response time control = 2731.82 ms; F(1, 35) = 7.35, $p = 0.011$, adj. $p = 0.032$).

3.3.2. Glass Pattern

Threshold

There was no significant difference between CVI and control groups in threshold for the glass pattern task (CVI $_{mean}$ = 0.52, 0.20 s.d., control $_{mean}$ = 0.45, 0.15 s.d.; t(36) = 1.12, $p = 0.27$, adj. $p = 0.81$). Adjusting for age did not impact the results (F(1, 37) = 1.10, $p = 0.30$, adj. $p = 0.90$).

Proportion Correct

There was no significant difference between CVI and control groups in proportion correct for the glass pattern task (CVI $_{mean}$ = 0.50, 0.15 s.d., control $_{mean}$ = 0.55, 0.11 s.d.; t(36) = −1.20, $p = 0.24$, adj. $p = 0.71$). Adjusting for age did not impact the results (F(1, 37) = 1.29, $p = 0.26$, adj. $p = 0.79$).

Response Time

There was no significant difference between CVI and control groups in response time for the glass pattern task (CVI $_{mean}$ = 2368.1 ms, 646.6 s.d., control $_{mean}$ = 2461.2 ms, 523.9 s.d.; t(36) = −0.48, $p = 0.63$). Adjusting for age did not impact the results (F(1, 37) = 0.21, $p = 0.65$, adj. $p = 1$).

*3.4. Strengths and Difficulties Questionnaire and Dutton Inventory*

Group Differences in SDQ and Dutton Outcomes Covarying for Age

The group of participants with CVI demonstrated a higher score on the emotional problems (F(1, 34) = 8.83, $p = 0.0056$, adj. $p = 0.045$), the peer problems scale (F(1, 34) = 4.40, $p = 0.044$, adj. $p = 0.35$), total difficulties score (F(1, 34) = 6.38, $p = 0.017$, adj. $p = 0.13$), and internalizing scores (F(1, 34) = 10.70, $p = 0.0026$, adj. $p = 0.021$) from the SDQ compared to controls. Of these, peer problems and total difficulties did not survive correction for multiple comparisons. There were no significant group differences for conduct problems (F(1, 35) = 1.50, $p = 0.23$, adj. $p = 1$), hyperactivity scale (F(1, 35) = 0.17, $p = 0.68$, adj. $p = 1$), prosocial scale (F(1, 35) = 1.11, $p = 0.30$, adj. $p = 1$), and externalizing scores

(F(1, 34) = 0.72, $p$ = 0.40, adj. $p$ = 1). There was no significant effect of age on any of these measures ($p > 0.05$).

Individuals with CVI had challenges on a significantly greater number of questions from the CVI inventory related to face recognition (F(1, 35) = 32.07, $p < 0.0001$, adj. $p < 0.0001$), object perception and recognition (F(1, 35) = 46.67, $p < 0.0001$, adj. $p < 0.0001$), and the five questions (F(1, 35) = 65.93, $p < 0.0001$, adj. $p < 0.0001$). There was no significant effect of age on any of these measures ($p > 0.05$).

The above analyses were repeated without adjusting for the potential impact of age. The results for the SDQ were slightly different, with significant increases in CVI for peer problems, emotional problems, and internalizing score, but not for the total difficulties score. The CVI inventory questions were virtually unchanged (see Supplementary Material for details).

*3.5. Partial Correlations between Behavioral Tasks and the SDQ*

As an exploratory analysis, a series of correlations between behavioral, SDQ, and CVI Inventory questions were performed. Across the whole sample, there were significant correlations between face threshold and the five questions (r = 0.35, $p$ = 0.045) and the objects questions (r = 0.37, $p$ = 0.032), with a trend for a positive correlation between face threshold and the faces questions (r = 0.33, $p$ = 0.062) and the peer problems scale (r = 0.33, $p$ = 0.068). There were also significant correlations between proportion correct on the face task and faces questions (r = −0.60, $p$ = 0.0002), object questions (r = −0.47, $p$ = 0.0064), and the five questions (r = −0.44, $p$ = 0.011). Moreover, significant Spearman correlations were observed between the faces questions and the peer problems score (r = 0.41, $p$ = 0.014), emotional problems scale (r = 0.51, $p$ = 0.0019), total difficulties score (r = 0.53, $p$ = 0.0011), and internalizing score (r = 0.60, $p$ = 0.0001).

## 4. Discussion

Overall, we observed a significant impairment in the performance of a face recognition task in our group of participants with CVI compared to controls, including an increase in threshold, reduction in the proportion correct, and an increase in response time. Similar impairments were not observed for the glass pattern task. Significantly higher scores on the SDQ were also observed in the CVI group compared to controls for emotional problems and internalizing scores after adjusting for the potential confounding effects of age. Finally, individuals with CVI also reported a greater number of difficulties on items from the CVI Inventory, specifically the five questions and those related to face and object recognition. These results indicate that CVI may be associated with significant impairments in face recognition that impact quality-of-life factors.

*4.1. Face Recognition Task*

Our results from the face recognition task are in line with previous reports of challenges recognizing faces in children with CVI. For example, Fazzi and colleagues investigated various cognitive visual profiles in children with CVI due to periventricular leukomalacia (PVL) and found that 86% demonstrated impaired visual recognition (as evaluated using color photographs of objects shown from unusual perspectives and lighting) and 18% demonstrated impairments in face recognition (as measured with the face memory subtest from the Test of Memory and Learning (TOMAL)) [29,30], suggesting that ventral stream impairments are relatively commonplace in patients with CVI. Moreover, children at risk of CVI due to preterm birth or low birth weight have also demonstrated impairments in face recognition [31,32], as well as emotion recognition [32]. Similarly, in our study, 16/19 (84.2%) of the respondents reported challenges (as indicated by a response of "sometimes" or "always") on at least one of the four face-related questions. Thus, along with the frequent parental reports of difficulty with face recognition, it is clear that face recognition needs to be more thoroughly investigated in this growing population of individuals, particularly given the important role that it plays in social and emotional development.

As highlighted in the literature, there are multiple factors that may contribute to difficulties with face recognition, including lighting, contrast, and the impact of positioning or facial expressions [33]. While here we controlled for some of these factors, what exactly is impaired within face recognition in CVI remains to be investigated. Moreover, the face recognition task implemented in this study utilized a Gaussian blur function, which may reflect Gestalt aspects of perception [34]. The blur function has also been well documented to impact the performance of recognition of unfamiliar faces [35–37]. However, in typically sighted individuals, the impact of blur on recognition can be in part mitigated by increasing the viewing time [38], as reflected in our data by the increased response time in both groups for the faces task as compared to the glass pattern task. Despite the increased time to examine the face stimuli, individuals with CVI still performed worse than the control group. While this remains to be fully appreciated, it is possible that by removing the detailed information of the target face, the participants needed to rely more on holistic face recognition processes. It has yet to be determined whether individuals with CVI demonstrate a selective impairment in holistic versus featural face recognition.

### 4.2. Glass Pattern

In this study we did not observe any significant differences between CVI and control groups on the glass pattern task. There are limited reports in the literature of the glass task being used to investigate form perception in CVI; however, increased thresholds (i.e., worse performance) have been noted in related neurodevelopmental disorders, such as autism spectrum disorder [39,40] and Williams syndrome [41,42]. This discrepancy in results may be in part due to the age of participants in our study. Specifically, while visual behaviors in other neurodevelopmental disorders are often investigated in children, our study focused on adolescents through middle-aged adults and thresholds are known to increase throughout the lifespan [43].

The null findings on the glass pattern task were also somewhat surprising as 15/19 (78.9%) of the respondents in this study reported challenges (as indicated by a response of "sometimes" or "always") on at least one of the five object-related questions. The inconsistency between self-report and quantitative measures observed in this study may be related to the glass pattern task itself, which fundamentally tests more mid-level visual perception [44,45], such as segmentation and integration based on spatial organization of black and white dots, whereas the questionnaire asks about identifying and naming objects themselves. In other words, the glass patterns may not be reflective of tangible objects and therefore may not represent the full spectrum of challenges with object recognition and identification that are reported in individuals with CVI [46,47]. Additional investigation is required to fully appreciate these differences.

### 4.3. Strengths and Difficulties Questionnaire

We observed significantly more difficulties in emotional problems and internalizing scores in the CVI group compared to controls. Similar findings have been reported in school-age children with congenital ocular visual impairment [48], as well as those in associated neurodevelopmental conditions including preterm birth and cerebral palsy [49–51]. Taken together, the evidence to date suggests that quality of life is impacted in individuals with CVI across the lifespan and strategies to ameliorate this in school, social, and workplace settings should be incorporated into (re)habilitation programs at all ages of CVI-onset.

### 4.4. Potential Impact of Age

Evidence suggests that the ventral stream continues to develop into adulthood, both in terms of behavioral performance [52,53] and the development of category-specific functional organization of the cortex [54]. The development of face and object recognition may also occur on different timescales. Specifically, although face and form perception begin during infancy [55–58], peak performance on object and form perception tasks, such as the glass pattern task, occurs at approximately 9 years of age [59], while peak performance on

face recognition tasks, such as the Cambridge Face Memory Task and the Old/New Faces Test, does not occur until roughly 31.5 years of age [52]. This corresponds to the timeline of the functional organization of the temporal cortex. The parahippocampal place area and lateral occipital cortex demonstrate selectivity for places and objects as early as 5 years of age, whereas selectivity of the fusiform face area, occipital face area, and superior temporal sulcus for face stimuli does not fully emerge until adulthood [54,60]. Nonetheless, our findings were robust to the potential effects of age for both the face recognition and glass pattern detection tasks.

This study has a few potential limitations. First, participants who volunteered to participate were aware that this study was focused specifically on face recognition. In other words, individuals with CVI and difficulty with face recognition may have been more likely to participate than those without difficulties with face recognition through self-selection. Second, participants in the CVI group self-identified as having CVI. Since this was a web-based study, there was no way for us to verify their diagnosis or determine the extent or impact of potential co-occurring ocular impairments. To this end, had visual acuity or contrast sensitivity been the driving factors behind the performance on the two tasks, we would have anticipated seeing poor performance on thresholds for both tasks. However, this was not what we observed, providing some level of confidence that the outcomes were not due to reductions in acuity or poor contrast sensitivity. Future investigations will need to be performed in person and with additional face image datasets [33] to verify the findings from this study.

*4.5. Implications for CVI*

The results from this study suggest that challenges with ventral stream visual processes, specifically face recognition, frequently occur in individuals with CVI. The results from the SDQ also indicate that internalizing behaviors, including emotional and social challenges, are higher in this same population; however, we were unable to observe significant correlations between face perception and SDQ outcomes in the CVI group. Nonetheless, based on the prosopagnosia literature, we know that challenges with face recognition have negative implications for social development, as well as skills such as joint attention, theory of mind, and other cognitive functions that are derived to some extent from face perception.

On a practical level, these results confirm that face recognition should be empirically evaluated in all children and adults with CVI, particularly as they may have developed compensatory strategies that enable them to identify close family, friends, and coworkers based on other non-facial features, such as voice, gait, or mannerisms.

The SDQ results from this study also highlight the need to increase awareness and evaluation of potential mental health issues and social–emotional factors in individuals with CVI.

**5. Conclusions**

In this online study, we observed a selective impairment in face recognition, but not form recognition, in participants with CVI compared to typically developing and sighted controls. Together with the existing body of literature, the evidence to date suggests that challenges with face recognition may be more prevalent than was once thought in individuals with CVI and that further investigation into this domain is necessary to more fully appreciate the depth and impact of difficulties with various aspects of face recognition and perception in this population of patients. The potential underlying neural mechanisms which may correspond to the selectivity of face recognition deficits in individuals with CVI have yet to be elucidated. The results from this study provide empirical data supporting the anecdotal reports of impaired face recognition in individuals with CVI. Thus, targeted evaluations of face recognition are warranted in individuals with CVI.

**Supplementary Materials:** The following supporting information can be downloaded at: https://www.mdpi.com/article/10.3390/vision7010009/s1, Additional Methods S1A: Selected Questions from the CVI Inventory; Additional Results S1B: Additional Statistical Analyses.

**Author Contributions:** Conceptualization, C.M.B., J.R., D.D.D. and P.J.B.; methodology, C.M.B., J.R., D.D.D. and P.J.B.; software, C.M.B., J.R., D.D.D. and P.J.B.; validation, C.M.B., J.R., D.D.D. and P.J.B.; formal analysis, C.M.B., H.C., D.D.D. and P.J.B.; investigation, C.M.B., C.E.M., J.R., D.D.D. and P.J.B.; resources, C.M.B., J.R., D.D.D. and P.J.B.; data curation, C.M.B., C.E.M., J.R. and P.J.B.; writing—original draft preparation C.M.B.; writing—review and editing, C.M.B., C.E.M., H.C., J.R., D.D.D. and P.J.B.; visualization, C.M.B. and P.J.B.; supervision, C.M.B., J.R., D.D.D. and P.J.B.; project administration, C.M.B., C.E.M., J.R., D.D.D. and P.J.B.; funding acquisition, C.M.B. and P.J.B. All authors have read and agreed to the published version of the manuscript.

**Funding:** This work was supported by R01 EY030877 for C.M.B. and P.B. was supported by R01 EY029713.

**Institutional Review Board Statement:** All subjects confirmed via checkbox that they had read and agreed to an online informed consent form approved by the Northeastern University Ethics Board IRB reference # 14-19-16 Psychophysical Study of Visual Perception and Eye Movement Control) before the experiment began. The experimental procedure was approved by the institutional review board at Northeastern University, and the experiments were performed in accordance with the tenets of the Declaration of Helsinki.

**Informed Consent Statement:** Informed consent was obtained from all subjects involved in the study.

**Data Availability Statement:** The data presented in this study are available on request from the corresponding author following approval of institutional data use agreements.

**Acknowledgments:** The authors wish to thank the individuals who chose to participate in this study, without whom this work would not be possible. We also thank the many individuals with CVI, as well as caregivers, educators, and healthcare providers of those with CVI who have diligently provided anecdotal reports of challenges with face recognition in this population of individuals. The development of the Interdisciplinary Affective Science Laboratory (IASLab) Face Set was supported by the National Institutes of Health Director's Pioneer Award (DP1OD003312) awarded to Lisa Feldman Barrett.

**Conflicts of Interest:** The authors declare no conflict of interest.

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
