# Peer review of "Deficits in Face Recognition and Consequent Quality-of-Life Factors in Individuals with Cerebral Visual Impairment"

_2411-5150, 2022_

Round 1

Reviewer 1 Report

This online study examined, in individuals with CVI (cerebral visual impairment), face recognition and form perception and their potential impact on social-emotional quality of life factors. The manuscript is well written and the statistical analysis appropriate. 

I only have a few relatively minor points for the authors to consider.

•     My pdf displayed a crowded Table 1 because it includes, perhaps unnecessarily, four decimal points. My suggestion is to present only two decimals throughout the paper and to explicitly state that p < 0.05 was chosen as the critical value for statistically significant effects.

•     Line 114: Replace ‘Gaussian’ with ‘Gaussian function’.

•     Line 175: Add ‘analyses’ so that the line reads: “... mixed model analyses with and without controlling for the potential effects of age.”

•     Lines 370 - 374: Consider breaking this long sentence into shorter ones.

•     Conclusion: I agree with the authors that glass patterns probably do not reflect real world objects and may not capture the difficulties in form recognition and identification that individuals with CVI often have. Other visual tasks involving crowding and object recognition in cluttered spaces may be better avenues for evaluating ventral stream dysfunction in CVI in the future.

Author Response

We wish to thank the reviewer for their helpful suggestions and feedback.

  • Table 1 has been updated to include only two significant digits where applicable. The column representing standard deviation was also combined with the mean columns so the data is now presented as: mean (standard deviation). This provided a little extra space and the table should now be more clearly viewed.
  • We added a sentence to the statistics section explicitly stating that p < 0.05 was considered as the critical value representing statistical significance. P-values throughout the text were also limited to 2 significant digits where applicable.
  • We added ‘function’ to line 114 and ‘analyses’ to line 175 as suggested.
  • Lines 370 – 374 were split into two sentences.

Reviewer 2 Report

-          The paper’s organization should be added in the last paragraph of the introduction.

-          For the legends of the figures, it is not acceptable to put screenshots!! So, legends should be rewritten.

-          The manuscript contains several mistakes and requires proofreading.

-          Some abbreviations are not defined in the manuscript. 

-          The manuscript requires serious revision and organization. For example, in Section 2.6, there are different styles of writing.

-          Page 8: what is the utility of the following sentence: “3.6. Non-repeated measures covarying for age”!!?

-          Authors should add and discuss some related works.

-          Add some perspectives in the last part of the conclusion.

-          The main contributions of the presented work should be highlighted in the abstract, introduction, and conclusion sections.

-          The below papers has some interesting implications that you could discuss in your Introduction and how it relates to your work: https://doi.org/10.3390/electronics9081188 and https://doi.org/10.1016/j.dsp.2020.102809

Author Response

We thank the reviewer for the insightful feedback and comments. We have addressed each of their concerns below.

  • The legends for the table and figures have been reformatted
  • We thank the reviewers for reporting the typos and formatting issues, which resulted during the transfer of the original file to the template. We have adjusted the text formatting of all section and sub-sections (such as “3.6 Non-repeated measures covarying for age”). These headings and sub-headings should now be more clearly differentiated from the main body of the text.
  • Regarding this comment from the reviewer: “Page 8: what is the utility of the following sentence: “3.6. Non-repeated measures covarying for age”!!?” This text represents a header and was formatted incorrectly once it was put into the MDPI template after the submission process. Thank you for pointing this out – the formatting has now been adjusted accordingly.
  • We added a definition of any abbreviations that were not previously described
  • Thank you for sharing the two review articles on computer-based face recognition. They indeed highlight the complexity of face recognition. This has been added to the discussion section.
  • Per the reviewer’s suggestion, we have included more discussion of related works and the potential implications for face recognition processes. (in particular, see lines 807 – 821)
  • The contributions of the present work are more clearly highlighted in the abstract, introduction, and conclusion sections
  • We have also bolstered the conclusion section with more implications for the assessment of this population of individuals.

Round 2

Reviewer 2 Report

The responses of the authors are convincing. 

The manuscript was improved compared to the initial version. 

For all these reasons, I recommend the acceptance of the manuscript.